# Eribulin and Paclitaxel Differentially Alter Extracellular Vesicles and Their Cargo from Triple-Negative Breast Cancer Cells

**DOI:** 10.3390/cancers13112783

**Published:** 2021-06-03

**Authors:** Petra J. Pederson, Huiyun Liang, Daria Filonov, Susan L. Mooberry

**Affiliations:** 1Department of Pharmacology, Mays Cancer Center, University of Texas Health Science Center at San Antonio, San Antonio, TX 78229-3900, USA; jans@livemail.uthscsa.edu (P.J.P.); liangh@uthscsa.edu (H.L.); 2Alpha Nano Tech, Morrisville, NC 27709, USA; daria@alphanano.tech

**Keywords:** extracellular vesicles, exosomes, microvesicles, microtubule targeting agents, eribulin, paclitaxel, vinorelbine, CD63, tetraspanin, triple-negative breast cancer

## Abstract

**Simple Summary:**

Cancer cells release small vesicles called extracellular vesicles (EVs) and these vesicles and their cargo promote cancer progression. The release of these vesicles is coordinated by cellular structures called microtubules. Drugs used in the treatment of breast cancer include eribulin and paclitaxel and they function by inhibiting microtubules. We investigated if these drugs would alter the release of vesicles from breast cancer cells and change their cargo. While our results show that these drugs do not alter the concentration of vesicles released, eribulin but not paclitaxel reduced levels of an important cargo called ILK. This is significant because ILK promotes cancer progression in breast cancer models. These results show that drugs can differently affect the release and cargo of extracellular vesicles and this could be significant for their clinical actions.

**Abstract:**

Extracellular vesicles play a central role in intercellular communication and contribute to cancer progression, including the epithelial-to-mesenchymal transition (EMT). Microtubule targeting agents (MTAs) including eribulin and paclitaxel continue to provide significant value in cancer therapy and their abilities to inhibit oncogenic signaling pathways, including eribulin’s capacity to reverse EMT are being revealed. Because microtubules are involved in the intracellular trafficking required for the formation and cargo loading of small extracellular vesicles (sEVs), we investigated whether MTA-mediated disruption of microtubule-dependent transport would impact sEV release and their cargo. Eribulin and paclitaxel caused an intracellular accumulation of CD63, a tetraspanin component of sEVs, in late/multivesicular endosomes of triple-negative breast cancer cells, consistent with the disruption of endosomal sorting and exosome cargo loading in these cells. While the concentrations of sEVs released from MTA-treated cells were not significantly altered, levels of CD63 and the CD63-associated cargos, ILK and β-integrin, were reduced in sEVs isolated from eribulin-treated HCC1937 cells as compared to vehicle or paclitaxel-treated cells. These results show that eribulin can reduce specific sEV cargos, including ILK, a major transducer of EMT in the tumor microenvironment, which may contribute to eribulin’s ability to reverse EMT to promote anticancer efficacy.

## 1. Introduction

Intercellular communication mediated by extracellular vesicles (EVs) is emerging as a pathway of central importance in both normal physiology and in the pathophysiology of disease [1,2,3]. EVs are widely implicated in multiple aspects of cancer progression [4], including the transfer of oncogenic proteins [5], angiogenesis [6,7,8], drug resistance [9], immune suppression [10,11], metabolic reprogramming [12,13], metastasis [14,15], and epithelial-to-mesenchymal transition (EMT) [16,17]. A recent study identified a central driver of EMT, integrin-linked kinase (ILK), as a cargo of CD63 positive (CD63+) EVs secreted from breast cancer cells [13]. These authors showed that EV-associated ILK is sufficient to promote EMT and metabolic reprogramming in recipient mammary epithelial cells, thus highlighting the key role of EV cargo in breast cancer progression [13].

Two main classes of EVs can be defined based on their biogenesis: exosomes and microvesicles. Exosomes are typically 50–150 nm in diameter and are formed by inward budding of late endosomes forming multivesicular endosomes (MVEs). MVEs are transported along the microtubules and can either be degraded upon fusion with a lysosome or their contents secreted by fusion with the plasma membrane [2,18]. Microvesicles range from 50–1000 nm in diameter and are formed by outward budding of the plasma membrane, requiring much of the same cellular machinery as the biogenesis of exosomes [2,19]. Due to the significant overlap in the size and cargos of microvesicles and exosomes, current methods of isolation cannot easily distinguish between them. The significant challenges in identifying how signaling molecules are selected, packaged, and secreted as exosome and microvesicle cargos and how this is regulated have been illuminated [20]. The term EVs is considered an accurate descriptor of the heterogeneous mixtures isolated by typical protocols [21]. In the present study, the term small EVs (sEVs) refers to EVs isolated from tissue culture media using a size-based method, differential centrifugation, to enrich for EVs < 200 nm.

Microtubule-dependent trafficking is involved in the secretion of exosomes formed through the MVE pathway and likely in the release of microvesicles as well [2,22]. Previous studies showed that the microtubule destabilizer nocodazole decreased the levels of CD63, a tetraspanin known as a characteristic sEV-associated protein, in EVs released from human kidney epithelial cells, highlighting the role of microtubules in EV secretion likely through the MVE pathway [23]. Another study showed that cold-induced microtubule depolymerization increased microvesicle release from the surface of the cancer cells, as demonstrated by the increased membrane vesicle shedding visualized microscopically, which was reversed by microtubule stabilization [22]. Together, these findings led to the hypothesis that microtubule-targeting agents (MTAs) used in the treatment of breast cancer would affect EV formation, release, and cargo.

There are two classes of MTAs characterized by their divergent effects on microtubules. Microtubule stabilizers increase the density of cellular microtubules, while microtubule destabilizers cause microtubule loss by promoting depolymerization [24]. Classically, MTAs have been considered solely as antimitotics due to their ability to induce abnormal mitotic spindles and mitotic arrest in rapidly dividing cancer cells in vitro. Accumulating evidence suggests however, that the non-mitotic effects of MTAs contribute significantly to their clinical efficacy [25,26,27]. Evidence for this includes the relatively slow rate of tumor doubling time in patients (45–325 days in breast cancer) [25], the low mitotic index in patient tumors (<1%) as compared to preclinical models [26], as well as in vivo microscopy experiments that have shown cells undergoing apoptosis after exposure to MTAs without ever entering mitosis [28,29,30]. MTAs interrupt key oncogenic signaling pathways by interfering with interphase protein trafficking, including the movement of transcription factors, for example, the androgen receptor [31], to the nucleus and the nuclear transport of DNA repair proteins after DNA damage [32,33]. The microtubule destabilizer, eribulin, has been shown to inhibit Smad2/3 nuclear trafficking and inhibit EMT-associated transcriptional activation that may contribute to its ability to reverse EMT in vitro, in preclinical models and in breast cancer patients [34,35,36]. These effects were not observed with the microtubule stabilizer paclitaxel, demonstrating that MTAs can have distinct microtubule-dependent consequences on oncogenic pathways that could be associated with their clinical efficacies. There is also evidence that effects on the tumor microenvironment are important predictors of response to MTAs, for example, in metastatic breast cancer patients treated with eribulin, those who had a decrease in circulating TGF-β post treatment had significantly improved progression-free survival [37]. Additionally, eribulin has been shown to remodel abnormal tumor vasculature in patients [38] and in preclinical models, where this effect correlated strongly with tumor regression [39].

Since EVs from cancer cells can impact oncogenesis and tumor progression and because the biogenesis of EVs is microtubule dependent, the goal of this study was to evaluate if eribulin or paclitaxel alters small EV (sEV) secretion and cargo. The results show that, indeed, disruption of cellular microtubules by MTAs used in the treatment of breast cancer affects the biogenesis and secretion of sEVs as well as specific sEV cargos, including ILK, with differences noted between microtubule stabilizers and microtubule destabilizers. Importantly, this work adds significant insights to our understanding of the complex effects of these cancer therapeutics on sEVs and the tumor microenvironment.

## 2. Materials and Methods

### 2.1. Cell Culture

MDA-MB-231 and HCC1937 triple-negative breast cancer (TNBC) cells were purchased from American Type Culture Collection (Manassas, VA, USA). Cell lines were validated by STR profiling by Genetica DNA Laboratories (Burlington, NC, USA). MDA-MB-231 cells were cultured in IMEM supplemented with 10% FBS and 30 µg/mL gentamicin, and HCC1937 cells were maintained in RPMI-1640 supplemented with 50 µg/mL gentamicin and 10% FBS. Cells were grown at 37 °C in 5% CO_2_ in a humidified environment. Paclitaxel (Sigma-Aldrich, St. Louis, MO, USA), eribulin (Eisai Inc., Woodcliff Lake, NJ, USA), and vinorelbine (AdooQ Biosciences, Irvine, CA, USA) were reconstituted in DMSO and stored at −20 °C.

### 2.2. Indirect Immunofluorescence

Approximately 50,000 cells per well were plated on glass coverslips in a 24-well plate and allowed to adhere overnight followed by treatment with the indicated concentrations of drugs for the indicated time. Cells were then fixed with 4% paraformaldehyde and permeabilized with 0.5% Triton X-100 in PBS and blocked with 10% bovine calf serum in PBS prior to immunostaining. The primary and secondary antibodies used, and their sources are provided in Appendix A. Images were acquired with a Nikon Eclipse 80i fluorescence microscope using a 60 × or 100 × objective. All images are focused stacks generated using NIS elements software. For high-content image analysis, 10,000 cells per well were plated on 96-well clear bottom cell-carrier plates (PerkinElmer, Waltham, MA, USA) and imaged at 20 × in a single focal plane with the Operetta™ (PerkinElmer). Columbus™ (PerkinElmer) software version 2.8.0 was used for the analysis of CD63 localization and intensity. Cell nuclei were defined by DAPI and cytoplasm defined by CellMask™ Blue (Thermo Fisher Scientific, Waltham, MA, USA) staining. CD63 intensity was separately calculated for the entire cell and for selected cell regions. The regions were defined using the Resize Region method with the outer and inner borders set to 0% and 35%, respectively, for the peripheral cytoplasm, and 35% and 55%, respectively, for the perinuclear region. Cellular CD63 spots were quantified using the Find Spots method C in the Columbus™ software version 2.8.0.

### 2.3. Cell Cycle Analysis

Flow cytometry was used to evaluate the effects of eribulin and paclitaxel on the cell cycle distribution of TNBC cells. Approximately 1 × 10^6^ cells per well were plated in 6-well plates, allowed to adhere overnight and then treated for 8 h in serum-free media, collected by scraping, and washed once with PBS. Single cell suspensions were achieved by passing cells through a cell strainer snap cap in a Falcon polystyrene test tube. Cells were then stained with Krishan’s reagent [40] and propidium iodide intensity measured on a Guava Muse® Cell Analyzer (EMD Millipore, Burlington, MA, USA) following the manufacturer’s protocol. FlowJo (Becton, Dickinson and Company, Ashland, OR, USA) was used for data analysis and calculation of cell cycle distribution and data were expressed as percentage of cells in the G_1_, S, and G_2_/M phases of the cell cycle based on DNA content.

### 2.4. Isolation of Small EVs

Small EVs (sEVs) are defined as EVs isolated from tissue culture media using a size-based method, differential centrifugation, to enrich for EVs < 200 nm. For each experiment, cells were grown to 80–90% confluency in four 300 cm^2^ tissue culture flasks per condition. Cells were then washed twice with PBS and replenished with 50 mL of serum-free media containing drug or vehicle for 8 h. Cell number and viability by trypan blue exclusion were determined at the time of conditioned media collection confirming totals of approximately 1.2 × 10^8^ (MDA-MB-231) or 7 × 10^7^ (HCC1937) viable cells per experimental condition. Approximately 1 × 10^7^ cells per condition were used for the preparation of whole cell lysates. Pooled conditioned media were first centrifuged at 500× *g* for 5 min to collect floating cells followed by a centrifugation of 1790× *g* for 10 min to remove dead cells and debris. Using a 70 Ti fixed angle rotor in a Beckman Coulter Optima L-90K ultracentrifuge, supernatants were centrifuged at 10,000× *g* for 30 min followed by ultracentrifugation at 120,000× *g* for 90 min to pellet sEVs. The sEV pellets were washed once with a small volume of PBS and pooled for a final 90 min spin at 120,000× *g* and the sEV pellets resuspended in 100 µL of PBS. An aliquot of 10–20 µL of sEV preparations was saved for nanoparticle tracking analysis. All sEV samples were stored at −80 °C for later analyses for no more than 4 months, and freeze thaw cycles were limited to once for NTA analyses and not more than about three times for Western blots.

### 2.5. Electron Microscopy

sEVs fixed in 2% paraformaldehyde were adsorbed onto formvar-carbon-coated EM grids (Electron Microscopy Sciences, Hatfield, PA, USA) and fixed with 1% glutaraldehyde, followed by eight washes with distilled water. Grids were then stained with 4% uranyl-oxalate (pH 7) and imaged on a JEOL JEM-1400 transmitting electron microscope at 80 kV with a magnification of 80,000×.

### 2.6. Nanoparticle Tracking Analysis (NTA)

NTA was conducted by Alpha Nano Tech, LLC (Morrisville, NC, USA) on the ZetaView Quatt (Particle Metrix, Meerbusch, Germany). On the day of analysis, DI water was filtered through a 0.22 µm syringe filter and its purity was confirmed by NTA prior to the study. Instrument calibration was performed with 100 nm polystyrene fluorescent beads (Fluorospheres, Lot # 2000230, Invitrogen, Carlsbad, CA, USA). Dilutions were made by mixing the DI water filtered through a 0.22-µm syringe filter with corresponding volume of a sample. The instrument settings used are provided in Appendix A.

### 2.7. Fluorescent Nanoparticle Tracking Analysis

Fluorescent NTA was conducted at Alpha Nano Tech, LLC on the ZetaView Quatt. The instrument settings used for scatter and fluorescent mode are provided in Appendix A. Anti CD63-AF488 antibodies (NBP2–42225AF488, Novus Biologicals, Littleton, CO, USA) were used at a stock concentration of 0.93 mg/mL. Five microliter of sEV sample and 1 μL of anti-CD63-AF488 antibodies were added to 14 μL of filtered PBS to achieve 1:20 working antibody dilution. The sample was incubated for 2 h at room temperature. After the 2-h incubation, 1.5 μL of labeled sEVs were added to 1.5 mL filtered DI water to achieve the final dilution 1:4000. Low bleach mode in Zetaview fNTA was used to minimize bleaching of the fluorescent dye. In this mode a small volume of sample is injected into the flow cell with the laser turned off before each video recording.

### 2.8. Immunoblotting

Whole cell lysates were prepared in cell extraction buffer (Invitrogen) supplemented with protease inhibitor cocktail (Sigma-Aldrich, St Louis, MO, USA), 1 mM PMSF, and 200 µM sodium orthovanadate. sEVs were lysed directly in NuPAGE™ LDS sample buffer (Thermo Fisher Scientific) and boiled prior to SDS PAGE. Non-reducing conditions were used for detection of CD63 and CD9. Equal amounts of protein (whole cell lysates) or equal volumes of sEVs (from equal numbers of cells) [41] were separated by SDS PAGE in Bolt 10% Bis-Tris Plus gels (Thermo Fisher Scientific) and transferred to Immobilon-FL PVDF membranes (EMD Millipore) for immunodetection of proteins. A list of antibodies is provided in Appendix A. Blots were probed with goat anti-mouse and anti-rabbit IRDye® secondary antibodies (LI-COR Biosciences, Lincoln, NE, USA) and imaged on an Odyssey FC (LI-COR).

### 2.9. Statistical Analyses

All statistical analyses were conducted using GraphPad Prism 8. One-way ANOVA with Tukey’s post-hoc test for multiple comparisons was used to compare the effects of vehicle, eribulin and paclitaxel treatment for high-content imaging, NTA, and Western blots. For comparisons between vehicle and vinorelbine an unpaired two-tailed t-test was used. Statistical significance was defined as a *p* value < 0.05.

## 3. Results

### 3.1. Eribulin and Paclitaxel Alter the Cellular Localization of CD63

MVEs, the intracellular precursors of exosomes, traffic along microtubules through the coordinated actions of Rab GTPases and microtubule motor proteins [42,43,44]. Because of the role of microtubules in MVE localization, we evaluated whether microtubule destabilization by eribulin or microtubule stabilization by paclitaxel would alter the localization of CD63, a tetraspanin membrane protein incorporated into MVEs and exosomes, in HCC1937 cells. We chose a short time course for these experiments because our prior studies have shown that eribulin-mediated microtubule depolymerization causes rapid changes in cell signaling events and that these effects can differ from those of microtubule stabilizers [36,45]. Within 2–4 h of MTA treatment at concentrations that disrupted interphase microtubules, changes in CD63 localization were apparent (Figure 1A and Appendix A) and differences between the drugs were noted. While perinuclear localization of CD63 was predominant in vehicle-treated cells, a higher density and larger CD63 spots were localized throughout the cytoplasm in eribulin-treated cells. In contrast, paclitaxel promoted CD63 staining that was not equally distributed throughout cells; it was often consolidated in discrete regions at the cell periphery (Figure 1A and Appendix A).

High-content imaging analysis and quantification demonstrated that total cellular CD63 intensity, as well as perinuclear and peripheral CD63 intensities were significantly increased with eribulin and paclitaxel as compared to vehicle-treated cells (Figure 1B and Appendix A). Additional analyses were conducted to more granularly identify and characterize CD63 spots. These analyses demonstrated that eribulin was distinct from paclitaxel in causing a significant increase in CD63 spot number and area (Figure 1C and Appendix A). Both paclitaxel and eribulin caused a statistically significant increase in total spot intensity per cell and the effect of eribulin was significantly higher than that of paclitaxel (Figure 1C). These data provided our first indication that eribulin and paclitaxel might differentially alter sEV biogenesis and cargo.

To identify if the eribulin- and paclitaxel-induced CD63 spots were localized at the late endosome, cells were co-stained for CD63 and RAB7. RAB7 is necessary for endocytic trafficking and maturation of early endosomes to late endosomes [46] and plays a role in the decision point between targeting MVEs to either the plasma membrane or the lysosome [42,47]. The results show that CD63 colocalized with RAB7 in both vehicle and MTA-treated cells (Appendix A). This is not surprising given that intraluminal budding and MVE formation are known to occur in late endosomes. However, the altered localization of RAB7 after short-term treatment with eribulin or paclitaxel as compared to vehicle-treated cells suggests that the MVE biogenesis pathway itself is affected by these drugs and that the differential localization of CD63 observed with eribulin or paclitaxel (Figure 1) is due to underlying differences in their effects on endosomal trafficking.

### 3.2. The Effects of Eribulin and Paclitaxel on sEV Release from TNBC Cells

To evaluate whether eribulin or paclitaxel alter sEV release, experimental conditions to isolate sEVs from the conditioned media of MTA-treated HCC1937 and MDA-MB-231 cells were investigated. While 50 and 500 nM concentrations of eribulin and paclitaxel respectively, over 2–4 h were adequate for evaluating the acute effects of microtubule disruption on CD63 localization, we determined the optimal drug treatment time and concentrations to maximize sEV collection yet limit the number of cells accumulated in mitosis. Concentrations as low as 25 nM eribulin or 50 nM paclitaxel were sufficient to promote altered CD63 localization (Appendix A). These concentrations did not cause substantial accumulation of cells in mitosis or initiate cell death over 8 h (Appendix A), yet still profoundly disrupted cellular microtubules within the much shorter, 2 h time frame (Appendix A). Of note, these drug concentrations are clinically relevant and well below the maximum plasma concentrations measured in patients, 2.7 µM for paclitaxel [48], and 505 nM for eribulin at the maximum-tolerated dose [49].

sEVs were isolated from the conditioned media of HCC1937 and MDA-MB-231 cells using differential centrifugation and the sEVs morphology and size were confirmed by electron microscopy (Figure 2A). The effects of eribulin and paclitaxel on sEV size and concentration were quantified using ZetaView™ nanoparticle tracking analysis (NTA). The majority of sEVs isolated from both HCC1937 and MDA-MB-231 cell conditioned media were between 50 and 200 nm in diameter and not changed by either drug (Figure 2B). In contrast to the effects of microtubule disruption on CD63 localization, surprisingly, we found no statistically significant differences in the mean concentration or median particle diameter of sEVs released by either cell line following treatment with eribulin or paclitaxel (Figure 2C,D).

Immunoblotting further confirmed that sEV-associated proteins, including the tetraspanins CD63, CD9 and CD81, and flotillin-1 were present in sEVs isolated from vehicle and MTA-treated TNBC cells, while GAPDH and the endoplasmic reticulum-associated protein, calnexin, were absent (Figure 3A,B and Appendix A). Interestingly, both eribulin and paclitaxel significantly decreased the CD63 content of sEVs from MDA-MB-231 (*p* = 0.0007 and 0.002, respectively) and HCC1937 cells (*p* = 0.0032 and 0.0396, respectively; Figure 3C and Appendix A), despite having had no significant effect on sEV concentrations measured by NTA. In contrast to the consistent effects of eribulin and paclitaxel in decreasing CD63 levels in sEVs from both cell lines, the effects of the two MTAs on CD9 were more variable. The CD9 content was significantly lower in sEVs from eribulin and paclitaxel-treated MDA-MB-231 cells (*p* < 0.0001), while in sEVs from HCC1937 cells, the CD9 levels were not significantly decreased by either drug (Figure 3D and Appendix A). The CD81 content was not significantly affected by MTA treatment in either cell line (Figure 3E and Appendix A). Finally, while the sEV flotillin content was not significantly changed by eribulin in either cell line, paclitaxel caused a statistically significant decrease in this protein in sEVs released from MDA-MB-231 cells (*p* = 0.0017), but not HCC1937 cells (Figure 3F and Appendix A).

The fact that of the three sEV-associated tetraspanins evaluated, only CD63 showed consistent and statistically significant decreases with both eribulin and paclitaxel in both cell lines raised the question of whether the cellular staining patterns of CD9 and CD81 were affected by MTAs. Consistent with earlier results (Appendix A), the intracellular localization of RAB7 and CD63 was disrupted by concentrations of MTAs used for sEV collection and they were colocalized (Appendix A). Comparatively, the localization of CD9 and CD81 was diffuse throughout the cytoplasm in all conditions, and this was unchanged by eribulin or paclitaxel (Appendix A). Together, these data show that while the size and concentration of sEVs released by MDA-MB-231 and HCC1937 cells was largely unchanged by eribulin or paclitaxel, the amount of CD63 was significantly reduced in these sEVs, possibly reflecting impaired loading of CD63 into exosomes at MVEs. Importantly this effect of MTAs was specific to CD63 and therefore does not reflect a non-specific reduction in sEV-associated proteins. These results suggest the intriguing possibility that the loading of CD63 into sEVs is regulated differently than the loading of CD9 and CD81 and that CD63 loading is more dependent on functional microtubules.

### 3.3. Eribulin and Paclitaxel Differently Alter the Release of Small CD63+ EVs

To specifically quantify the release of CD63+ sEVs, fluorescent nanoparticle tracking analysis (fNTA) using a CD63 antibody was employed. The effects of eribulin and paclitaxel on CD63+ sEV concentration and size were first evaluated in MDA-MB-231 cells (Figure 4A–D). The plot of concentration vs. size of the CD63+ population of sEVs from vehicle and paclitaxel-treated cells overlapped completely, with no differences in the size distribution (Figure 4A). In marked contrast, sEVs from eribulin-treated cells showed a completely different profile, with approximately 72% fewer CD63+ sEVs as compared to the vehicle control (Figure 4A,B). This eribulin-mediated decrease in the concentration of CD63+ sEVs (Figure 4B) was statistically significant when compared to either vehicle or paclitaxel treatment (*p* = 0.0114 and 0.0110, respectively). Moreover, while the percentage of CD63+ sEVs (of total sEVs) from vehicle or paclitaxel-treated cells averaged 58.4% and 58.8%, respectively, the percent CD63+ sEVs from eribulin-treated cells was significantly lower, 24%, as compared to vehicle or paclitaxel treatment (*p* = 0.0022 and 0.0014, respectively; Figure 4C). Thus, the lower concentration of CD63+ sEVs seen with eribulin is not because there are fewer total sEVs but is due to a specific decrease in CD63+ vesicles released from MDA-MB-231 cells. In contrast, both the concentration and percentage of CD63+ sEVs from paclitaxel-treated cells as measured by fNTA were identical to those of the vehicle control.

Consistent with the similar size distributions of sEVs from vehicle and paclitaxel-treated cells in the CD63+ populations (Figure 4A) there was also no difference in median particle diameter (Figure 4D). In contrast, the profile of CD63+ sEVs released from eribulin-treated cells showed a broadening of the size distribution with no apparent peak, but a plateau from approximately 100–225 nm (Figure 4A). Furthermore, there was a significant shift in median particle size of CD63+ sEVs from 120.5 nm, in vehicle controls, to 191.2 nm (*p* < 0.0001) from eribulin-treated cells (Figure 4D). The increase in median sEV size with eribulin, together with the marked change in CD63+ sEVs size distribution, strongly suggests that eribulin specifically alters the biogenesis of CD63+ sEVs in MDA-MB-231 cells.

The effects of eribulin and paclitaxel on the concentration and size of CD63+ sEVs from HCC1937 cells were also evaluated (Figure 4E–H). Similar to the effects seen in MDA-MB-231 cells, paclitaxel did not change the shape of the concentration vs. size graph of the CD63+ sEV population or the concentration of CD63+ sEVs (Figure 4E,F). In contrast to results in MDA-MB-231 cells, eribulin did not change the CD63+ sEV concentration vs. size profile or the total concentration of CD63+ sEVs as compared to vehicle-treated controls (Figure 4E,F). Nevertheless, and like the results in MDA-MB-231 cells, eribulin did cause a statistically significant decrease in the percentage of sEVs that were CD63+ (Figure 4G). Thus, in vehicle and paclitaxel-treated cells, 54.4% and 48.9% of the sEVs were positive for CD63, respectively, while only 37.5% of sEVs were CD63+ from eribulin-treated cells (*p* = 0.0042 and 0.0372 respectively) (Figure 4G). The size of the CD63+ sEVs was also evaluated, and the median diameters were similar for all treatments (Figure 4H). While there were some differences between the cell lines, taken together, the data show that in both cell lines, eribulin consistently decreased the percentage of sEVs that were CD63+, effects that were not seen with paclitaxel.

### 3.4. Vinorelbine Inhibits the Release of CD63+ sEVs from MDA-MB-231 Cells

To determine if the intriguing actions of eribulin on CD63+ sEVs in MDA-MB-231 cells are unique to this drug or generalizable to other microtubule depolymerizers, the effects of vinorelbine were studied. Like eribulin, a 4-h treatment with vinorelbine caused microtubule depolymerization and the accumulation of CD63 spots as visualized by indirect immunofluorescence (Appendix A). High-content imaging revealed significantly higher intensities of CD63 in the total cytoplasm, perinuclear and peripheral regions of vinorelbine-treated cells (Appendix A), as well as significantly more CD63 spots per cell, spot area per cell, and spot intensity as compared to vehicle-treated controls (Appendix A). These results show that microtubule depolymerization initiated by either vinorelbine or eribulin caused the same phenotype of CD63 localization and staining intensity. Consistent with this, we expected that vinorelbine would inhibit the formation and release of CD63+ vesicles from MDA-MB-231 cells.

sEVs were isolated from the conditioned media of cells treated with 25 nM vinorelbine or vehicle for 8 h, and morphology and size confirmed by electron microscopy (Appendix A). Immunoblot analysis of CD63, CD9, CD81, and flotillin-1 showed significant reductions in CD63 (*p* = 0.0024) and CD9 (*p* = 0.0040) but no effect on CD81 or flotillin-1 levels after vinorelbine treatment (Figure 5A,B and Appendix A), consistent with the effects of eribulin (Figure 3). The shape of the concentration vs. size profile for total sEVs (Figure 5C) showed a significant decrease in the overall sEV concentration with vinorelbine treatment (*p* = 0.0174) (Figure 5E). The size distribution of the total sEV population (Figure 5C), and median particle diameter (Figure 5H) were not changed by vinorelbine. The effects of vinorelbine on the total sEVs are consistent with the effects of eribulin, with the exception that vinorelbine decreased sEV concentration. Similar to the effects of eribulin, the concentration of CD63+ sEVs was decreased significantly, by 78%, with vinorelbine (*p* = 0.0195) (Figure 5D,F). Additionally, the percentage of CD63+ sEVs was substantially reduced by vinorelbine, from 52.3% of the sEVs from vehicle controls to 17.2% of the sEV population from vinorelbine-treated cells (*p* = 0.0256) (Figure 5G). Vinorelbine increased the median size of CD63+ sEVs from 125.3 to 155.9 nm (Figure 5I), but this increase did not reach statistical significance (*p* = 0.0886), in contrast to the effects of eribulin. Overall, these results suggest that the biogenesis of CD63+ sEVs, likely formed through the MVE pathway, are dependent on functional microtubules in MDA-MB-231 cells and while there are minor differences between eribulin and vinorelbine, their effects are remarkedly similar.

### 3.5. Microtubule Targeting Agents Differentially Affect ILK, an EV Cargo Implicated in EMT

Integrin-linked kinase (ILK) is a signaling protein that was recently shown to be an EV cargo upregulated in sEVs released from hypoxic estrogen receptor positive breast cancer cells [13]. In that model, ILK was specifically associated with CD63+ sEVs and initiated metabolic reprogramming related to EMT in recipient breast epithelial cells [13]. Based on the close association between CD63 and ILK, we hypothesized that the specific depletion of CD63+ sEVs by eribulin would also lead to decreased ILK sEV cargo. Indeed, in HCC1937 cells, eribulin significantly decreased ILK in sEVs relative to vehicle and paclitaxel-treated cells (*p* = 0.0042 and 0.0475 respectively) (Figure 6A,B and Appendix A), consistent with eribulin-mediated reductions in CD63 sEV content (Figure 3B,C). Although paclitaxel modestly decreased ILK levels, the difference was not statistically significant (*p* = 0.1432) (Figure 6A,B). While ILK is expressed abundantly in MDA-MB-231 cell lysates, the low levels of ILK in sEVs from MDA-MB-231 cells precluded evaluation of the effects of MTAs on sEV ILK content (Appendix A).

The link between CD63 and ILK loading into sEVs could be related to the fact that CD63 and other tetraspanins are known to physically interact with integrins in tetraspanin-enriched microdomains [50,51], and as its name implies, ILK is known for its colocalization with the cytoplasmic tail of β1-integrin [52]. Accordingly, we addressed whether eribulin or paclitaxel impacts the loading of β1-integrin into sEVs. In parallel with the effects on CD63 and ILK, eribulin decreased the levels of β1-integrin in sEVs from HCC1937 cells (*p =* 0.0311) (Figure 6B). Paclitaxel showed a similar trend toward decreased β1-integrin in sEVs from HCC1937 cells (*p =* 0.2734 (Figure 6B). Together, these results show that the loading of CD63-associated sEV cargos, ILK and β1-integrin are inhibited by eribulin-mediated microtubule depolymerization, but not by paclitaxel-induced microtubule stabilization.

## 4. Discussion

The goal of this work was to test the hypothesis that MTA treatment of TNBC cells would alter the release of sEVs and change their bioactive cargo. The expectation that MTAs would change the release and cargo of sEVs is based on the critical role of normal microtubule dynamics in multiple aspects of intracellular trafficking implicated in EV biogenesis, including MVE trafficking along microtubules [2] and microvesicle release [22,53]. The effects of MTAs on the localization of CD63 and RAB7 that were seen within 2–4 h, suggest that late endosome/MVE trafficking and the coordinated packaging of exosome cargo is indeed affected by MTAs. Furthermore, differences in the effects of eribulin and vinorelbine compared to paclitaxel, which caused a strikingly different pattern of CD63 staining as defined by the spot number, intensity, and area suggest that microtubule destabilizers and stabilizers disrupt sEV biogenesis pathways differently. While MTAs are often considered a single class of drugs, these results highlight differences between destabilizers and stabilizers and are consistent with their divergent effects on intracellular signaling including inhibition of TGF-β-induced nuclear translocation of Smad2/3 [36], and mitochondrial biogenesis and polarization [54,55].

In contrast to our initial expectations based on the CD63 phenotype visualized by immunofluorescence, sEV concentration and size measured by NTA showed that none of the MTAs significantly affected these parameters in sEVs released from HCC1937 cells and only vinorelbine significantly decreased the concentration of sEVs released from MDA-MB-231 cells. Thus, and perhaps surprisingly, gross suppression or induction of sEV biogenesis did not occur within an 8-h treatment of cells with eribulin or paclitaxel despite major disruption of cellular microtubules. These data contrast those of others who reported increased EV secretion by cancer cells treated with paclitaxel, in particular [56,57]. However, our approach differs from these groups in that we utilized an 8-h incubation time as opposed to their much longer exposures (up to 72 h), to separate the rapid microtubule-dependent effects of these drugs from their downstream consequences including initiation of mitotic arrest and apoptosis. The release of apoptotic bodies would confound the measurements of sEVs generated by either the MVE/exosome or microvesicle biogenesis pathways. Additionally, our focus on the non-mitotic effects of MTAs seen at early time points is important because patient data demonstrate that mitosis is a much rarer occurrence in breast tumors in patients than in cell culture models [25,27], and therefore, the non-mitotic effects of MTAs are likely to be more clinically relevant.

The lower levels of CD63 in sEVs from MTA-treated cells in both TNBC cell lines, together with the accumulation and altered localization of CD63 visualized by immunofluorescence, confirm that microtubules are critical for CD63 loading into sEVs. Moreover, our observations that the effects are more pronounced by microtubule loss than microtubule stabilization suggest that paclitaxel-stabilized microtubules retain some functionality for loading CD63. Our studies deciphering individual CD63+ sEVs, from the total sEV population showed that this is indeed the case. Eribulin and vinorelbine, but not paclitaxel, inhibited the release of CD63+ sEVs from MDA-MB-231 cells. In these cells, the substantial decrease in both the number and percentage of CD63+ sEVs with eribulin and vinorelbine demonstrates the critical role of microtubules in CD63 loading into sEVs. The importance of microtubules in loading CD63 into vesicles was also demonstrated in HCC1937 cells where eribulin caused a significant decrease in the percentage of CD63+ sEVs. Together, these data point to significant changes in sEV biogenesis and cargo loading initiated by microtubule depolymerization.

In addition to CD63 loading, eribulin caused a significant increase in the median size of CD63+ sEVs released from MDA-MB-231 cells. This may be due to an increase in sEV formation through the microvesicle pathway, which produces sEVs that are typically larger than exosomes released by the MVE pathway. This is consistent with prior work which showed that cold-induced microtubule depolymerization enhanced the release of microvesicles from cancer cells [22].

It is interesting to note that while paclitaxel, eribulin, and vinorelbine decreased the levels of CD63 in sEVs as measured by immunoblotting, paclitaxel had no effect on the concentration of CD63+ particles as measured by fNTA in both TNBC cell lines. We postulate that this is due to the technical differences between immunoblotting and fNTA. While immunoblotting provides information about the amount of CD63 in the sample on a continuum, fNTA evaluates individual particles and answers the binary question of whether or not they contain CD63 based on a threshold level of fluorescence. Thus, these differences between the Western blot and fNTA data suggest that paclitaxel decreases the amount of CD63 in the EV population, but it does not decrease the surface-accessible CD63 below the threshold detected by fNTA. Ultimately, the fNTA data suggest that the different effects of microtubule depolymerizers and stabilizers on the intracellular localization of CD63 also lead to diverging effects on CD63 loading into sEVs.

While the tetraspanins CD63, CD9, and CD81 are all implicated in sorting exosome cargo [2], the diverging effects of MTAs on these sEV tetraspanins also indicate that their transport and loading are not equally dependent on microtubules and that there are differences between cell lines. A similar effect was noted for flotillin-1, which was decreased by paclitaxel in MDA-MB-231 cells but not changed by any MTA in HCC1937 cells. These differences in sEV cargos initiated by MTAs argue against MTAs having a non-specific inhibitory effect on sEV protein loading and suggest a precise and cell-line-dependent role of microtubules in the transport and loading of CD63, CD9, flotillin-1, and other EV cargos.

Our finding that ILK levels are significantly lower in sEVs from eribulin-treated HCC1937 cells compared to vehicle or paclitaxel-treated cells is notable given the ability of sEV-associated ILK to promote an EMT phenotype in mammary epithelial cells [13]. Furthermore, the activity of ILK has been linked to EMT and breast cancer progression [58]. Additionally, preclinical and clinical studies demonstrate that eribulin can reverse EMT [34,35,59] and inhibit key EMT transcription factors [36]. Our study presents a mechanism by which eribulin could reverse EMT, by altering bioactive sEV cargo. Additional studies are warranted to determine if sEVs from eribulin-treated TNBC cells diminish EMT in recipient cells and how this compares to sEVs from paclitaxel-treated cells. Future studies are needed to identify additional TNBC cell lines where ILK is highly expressed in sEVs and if it is altered by MTAs. Important follow-up studies should evaluate the effects of MTAs on ILK levels in sEVs derived from patient serum. 

Intriguing differences were noted between the effects of paclitaxel as compared to eribulin and vinorelbine on CD63+ sEVs and on ILK and β-integrin cargo. It is interesting that others have shown that paclitaxel promotes the release of EVs from mammary tumor cells in vivo with pro-metastatic cargo [56]. Additionally, paclitaxel and nocodozale increased survivin, a protein implicated in chemoresistance, in EVs from MDA-MB-231 cells [57]. Recent studies show that paclitaxel and doxorubicin chemotherapy increased the levels of EV miRNAs associated with chemoresistance in patients and in preclinical models [60]. These results show that paclitaxel can impact EV cargo that may contribute to drug resistance. Further studies evaluating the effects of microtubule depolymerizing drugs such as eribulin and vinorelbine in comparison with paclitaxel are needed to fully understand the differences among MTAs in their ability to impact the tumor microenvironment to reverse EMT and/or promote stemness and drug resistance pathways.

In conclusion, our studies begin to identify that sEV cargos can be differentially altered by specific MTAs. Because sEVs represent an important mechanism of intercellular communication in the progression of cancer, the ability of MTAs to alter bioactive sEV cargo could have clinical implications both during and after treatment. Furthermore, the accessibility of circulating sEVs derived from patients’ serum makes them ideal biomarkers for the measurement of these cargos as dynamic readouts of drug response. The continued investigation of the effects of diverse MTAs on sEV cargo and function in breast cancer is a worthwhile endeavor that could reveal the full range of effects of these drugs in patients and ultimately inform on their optimal use in the treatment of breast cancer.

## 5. Conclusions

EVs are important mediators of intercellular communication in the tumor microenvironment and have been shown to impact cancer progression. We tested the hypothesis that MTAs used in the treatment of TNBC could alter the biogenesis and composition of sEVs released by TNBC cells. Our results show that eribulin and paclitaxel rapidly alter the trafficking of CD63 and MVEs and do so differently, leading to differential effects on the CD63 content of sEVs. Notably, the effects of these drugs on CD63 were specific in that the changes did not generalize to all sEV-associated proteins. Furthermore, we show that the CD63-associated cargo, ILK, which has been reported to promote an EMT phenotype in recipient cells, was reduced in sEVs by eribulin but not paclitaxel. Thus, our results support the conclusion that MTAs could differentially impact the tumor microenvironment through modulation of sEV cargo, which could have implications for their anticancer efficacy.

## Figures and Tables

**Figure 1 cancers-13-02783-f001:**
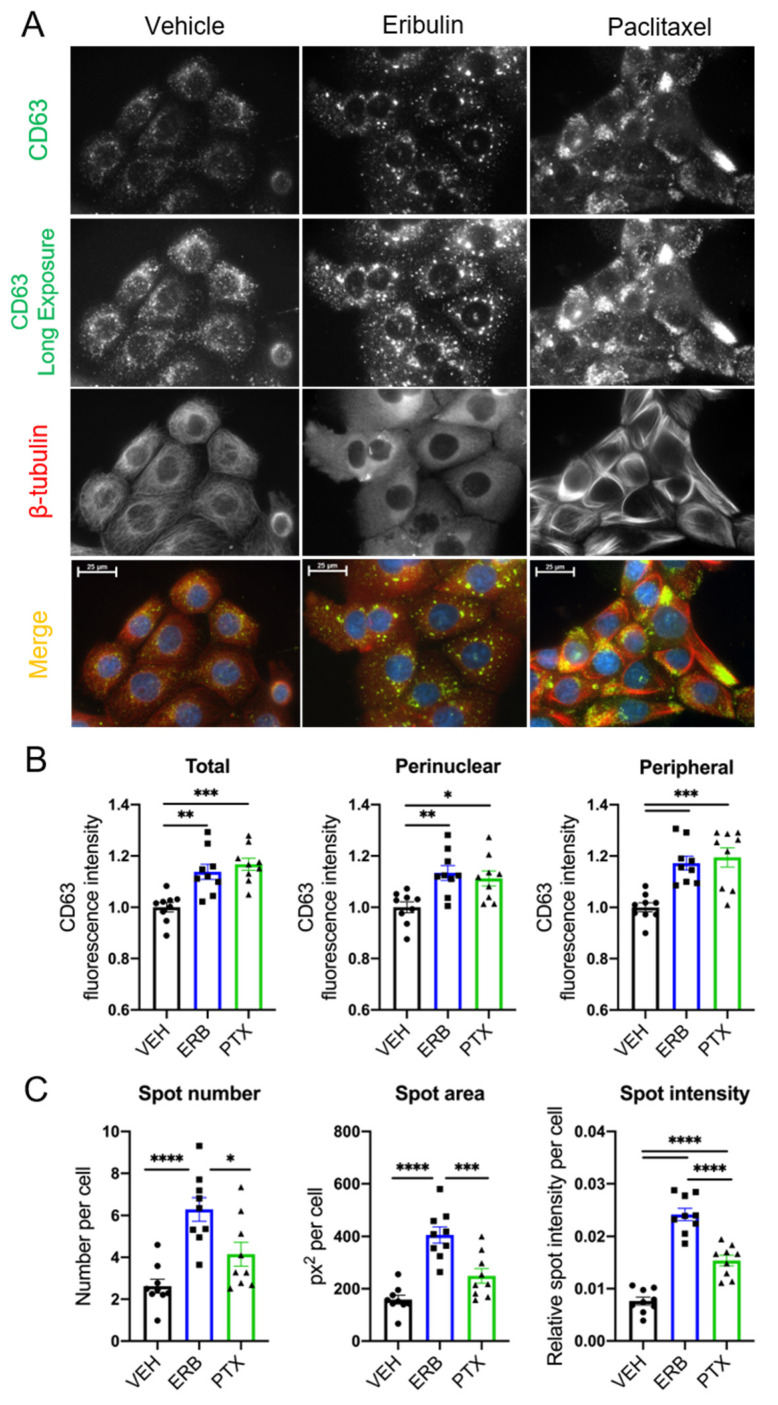
Effects of eribulin or paclitaxel on CD63 localization in HCC1937 cells. HCC1937 cells were treated with vehicle (VEH), 50 nM eribulin (ERB), or 500 nM paclitaxel (PTX) for 4 h to cause maximal microtubule disruption and then co-immunostained for CD63 (green) and β-tubulin (red). (**A**) Immunofluorescence images are representative of two experiments with equal exposures across conditions. Scale bar = 25 µm. (**B**) Quantification of CD63 regional intensity and (**C**) CD63 spots analysis using Operetta™ high-content imaging. Mean cellular fluorescence for a mean of 432 cells (range: 109–1265) was calculated for *n* = 9 wells and represents three independent experiments. Significance determined by one-way ANOVA with Tukey’s post-hoc test. * *p* < 0.05, ** *p* < 0.01, *** *p* < 0.001, **** *p* < 0.0001.

**Figure 2 cancers-13-02783-f002:**
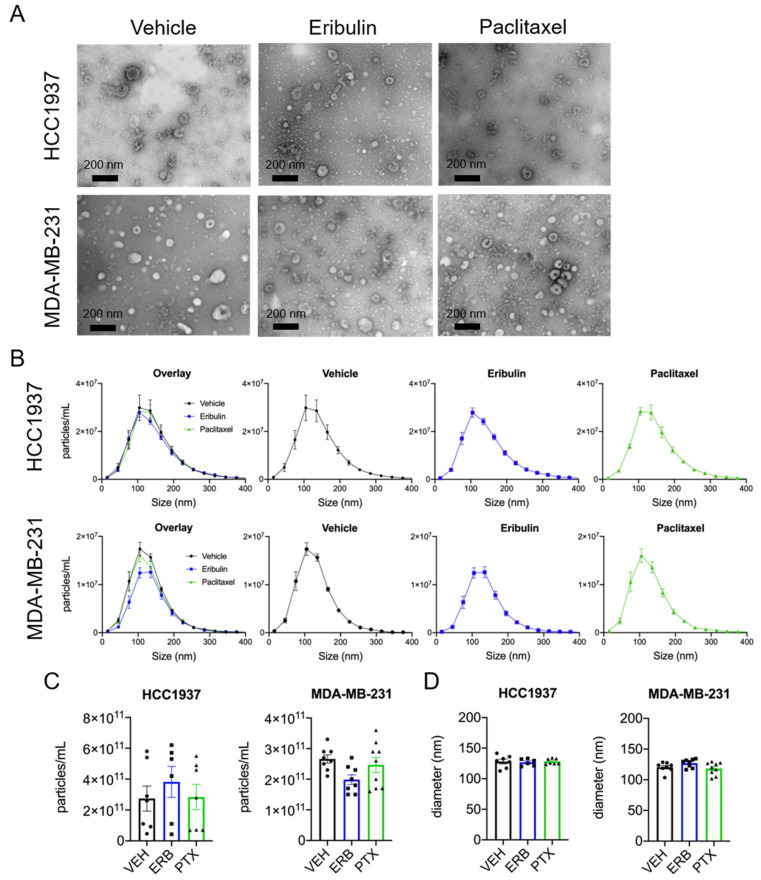
Characterization of sEVs by electron microscopy and NTA. Cells were treated for 8 h with vehicle, 25 nM eribulin, or 50 nM paclitaxel and sEVs collected from the conditioned media. (**A**) sEVs were evaluated by transmitting electron microscopy. Scale bar = 200 nm. Pictures are representative of three experiments. (**B**) Size distribution of sEVs determined by ZetaView™, means ± SEM are shown, *n* = 6–9. (**C**) Mean particle concentration determined by ZetaView™. (**D**) Median particle diameter determined by ZetaView™. Points represent the results of independent experiments and bars represent the mean ± SEM, *n* = 6–9. Significance determined by one-way ANOVA with Tukey’s post-hoc test, and no treatments were significantly different from the other.

**Figure 3 cancers-13-02783-f003:**
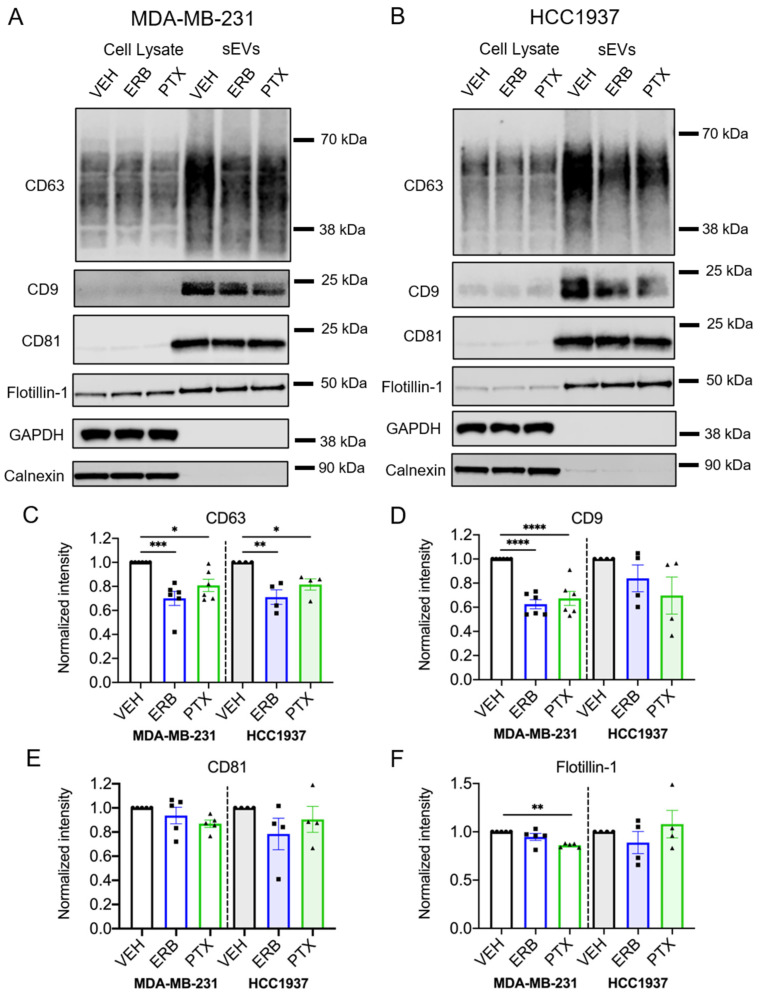
Effects of eribulin and paclitaxel on sEV-associated proteins. MDA-MB-231 and HCC1937 cells were treated for 8 h with vehicle, 25 nM eribulin, or 50 nM paclitaxel and sEVs were isolated from the conditioned media. (**A**,**B**) Representative immunoblots of indicated proteins in 10 µg of cell lysates and equal volumes of sEVs representing 7.5% (CD63, CD9, GAPDH) or 42% (calnexin, CD81, flotillin-1) of the yield of 1.2 × 10^8^ MDA-MB-231 cells (**A**) or 6–8 × 10^7^ HCC1937 cells (**B**). (**C**–**F**) Band intensities for indicated proteins in sEV samples normalized to vehicle control sEVs. Points represent the results of independent experiments and bars represent the mean ± SEM, *n* = 4–6. Significance determined by one-way ANOVA with Tukey’s post-hoc. * *p* < 0.05, ** *p* < 0.01, *** *p* < 0.001, **** *p* < 0.0001.

**Figure 4 cancers-13-02783-f004:**
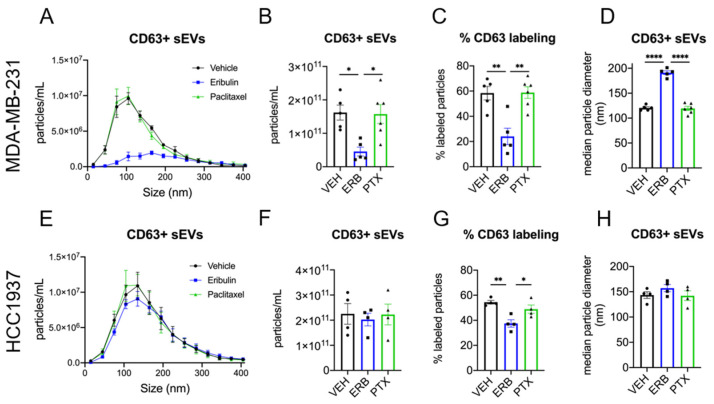
Fluorescent nanoparticle tracking analysis of CD63 positive sEVs. MDA-MB-231 (**A**–**D**) or HCC1937 (**E**–**H**) cells were treated for 8 h with vehicle, 25 nM eribulin, or 50 nM paclitaxel and sEVs collected from the conditioned media. Samples were labeled with anti-CD63-AF488 fluorescent antibodies to quantify CD63+ sEVs and NTA performed with the ZetaView™ in fluorescent mode to measure CD63+ particles. (**A**,**E**) Overlay of NTA size distribution vs. concentration graphs for each treatment measuring CD63+ sEVs. (**B**,**F**) The mean CD63+ particle concentration, (**C**,**G**) percent CD63+ particles, determined by dividing the CD63+ concentration by the total sEV concentration for each sample, and (**D**,**H**) median CD63+ sEV diameters for each treatment are shown. Points in the bar graphs represent the results of independent experiments and the bars represent the mean ± SEM, *n* = 4–6, and significance determined by one-way ANOVA with Tukey’s post-hoc test, * *p* < 0.05, ** *p* < 0.01, **** *p* < 0.0001.

**Figure 5 cancers-13-02783-f005:**
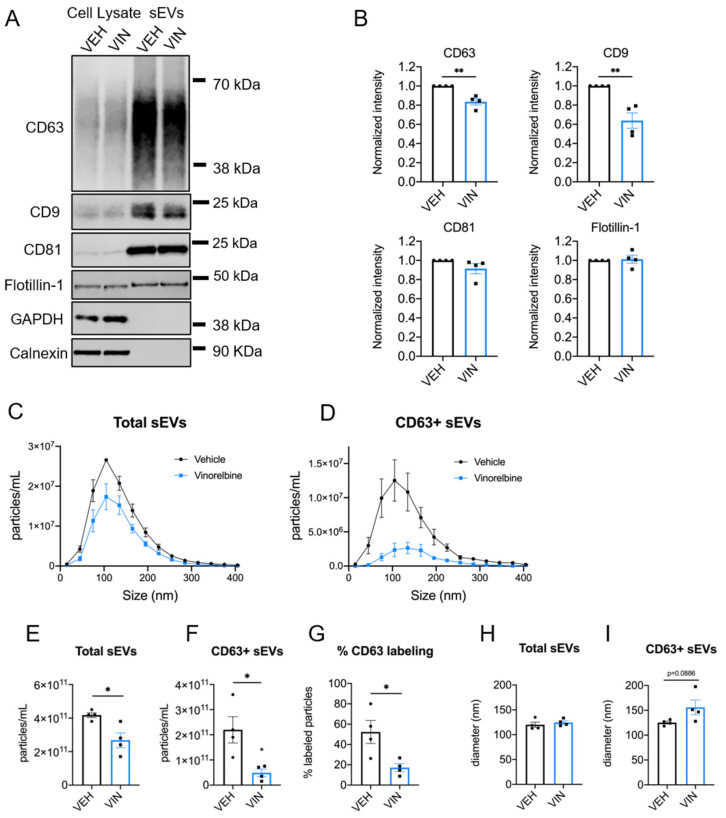
Effects of vinorelbine on sEVs released by MDA-MB-231 cells. Cells were treated for 8 h with vehicle or 25 nM vinorelbine and sEVs collected. (**A**) Representative immunoblots of indicated proteins in 10 µg of cell lysates and equal volumes of sEVs representing 7.5% (CD63, CD9, GAPDH), or 42% (calnexin, CD81, flotillin-1) of the yield of 1.2 × 10^8^ MDA-MB-231 cells. (**B**) sEV CD63 signal intensities from the immunoblots were quantified and normalized to vehicle control sEVs for each experiment. Graphs show the mean ± SEM for the relative levels of CD63. *n* = 4. (**C**,**D**) sEV samples were labeled with anti-CD63-AF488 fluorescent antibodies and NTA performed with ZetaView in scatter mode to measure total sEVs and in fluorescent mode to measure only CD63+ sEVs. Graphs show overlays of NTA size distribution curves for each treatment condition in total sEVs (**C**) and CD63+ sEVs (**D**). (**E**–**G**) The mean particle concentrations in total (**E**) and CD63+ sEVs (**F**) and the percent CD63+ particles (**G**), determined by dividing the CD63+ concentration by the total concentration for each sample. (**H**,**I**) Median particle diameters for each treatment of total (**H**) and CD63+ sEVs (**I**). Points in the bar graphs represent the results of independent experiments and bars represent the mean of these experiments ± SEM, *n* = 4, and significance determined by unpaired *t*-test, * *p* < 0.05, ** *p* < 0.01.

**Figure 6 cancers-13-02783-f006:**
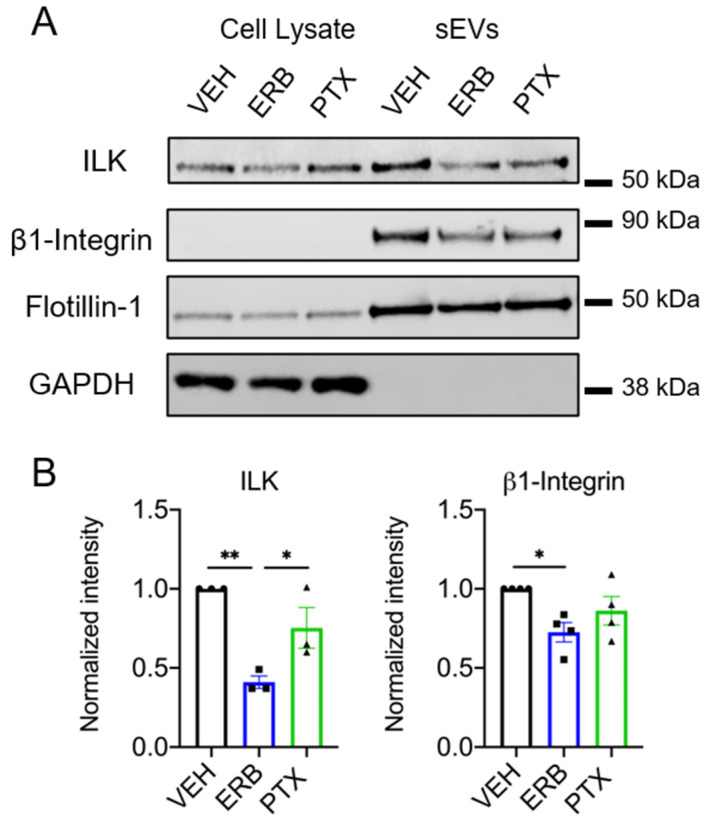
Effects of eribulin or paclitaxel on sEV-associated ILK and β1-integrin. HCC1937 cells were treated for 8 h with vehicle, 25 nM eribulin, or 50 nM paclitaxel and sEVs were isolated from the conditioned media. (**A**) Representative immunoblots of the indicated proteins in 10 µg of cell lysates and equal volumes of sEVs representing 42% (ILK, β1-integrin, flotillin-1) or 7.5% (GAPDH) of the yield from 6–8 × 10^7^ HCC1937 cells. (**B**) Band intensities for indicated proteins in sEV samples normalized to vehicle control sEVs. Points represent the results of independent experiments and bars the mean ± SEM, *n* = 3–4. Significance was determined among sEV samples by one-way ANOVA with Tukey’s post-hoc test, * *p* < 0.05 and ** *p* < 0.01.

## Data Availability

The data presented in this study are available on request from the corresponding author.

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
