# Peer review of "Eribulin and Paclitaxel Differentially Alter Extracellular Vesicles and Their Cargo from Triple-Negative Breast Cancer Cells"

_cancers, 2021, doi:10.3390/cancers13112783_

Round 1
Reviewer 1 Report
This manuscript by Pedersen et al. describes the effects of eribulin(MT depoymerizers) and paclitaxel (polymerized MT stabilizer) on the EV characteristics from MDA-MB-231 and HCC1937 TNBC cell lines. Both drugs modify the peripherilization of CD63+ spots in both cell lines. Although EVs aren’t modified, both decrease the CD63 content of EVs in both cell lines. Further, CD63+ EVs are reduced by eribulin but not paclitaxel, particularly in MB-231 cells. Vinorelbine shows effects similar to eribulin in 231 cells, suggesting this effect is specific to depolymerization. IN HCC1937 cells, the ILK and beta1-integrins level in EVs is decreased markedly with eribulin, and perhaps modestly with paclitaxel. The manuscript concludes that:
- Eribulin can reduce specific sEV cargos, including ILK (abstract)
- Disruption of microtubules affects the biogenesis and secretion of sEV and their cargos, including ILK with differences between microtubule stabilizers/destabilizers (introduction)
- sEV cargos can be differentially altered by specific MTAs (discussion)
- Eribulin and paclitaxel rapidly alter trafficking of CD63 and MVEs and do so differently.
The strengths of this manuscript is it directly compares the effects of different MTAs on EVs and their contents, using distinct methods including immunoblotting, EM, and sEV using Alpha Nano Tech instruments. The data shown largely support the stated conclusions, with the caveats. Some of the immunoblots, particularly in Figure 3, appear very well done. As noted below, the principle weaknesses concern the lack of demonstrated biological impact of the EV alterations, the discordant data for paclitaxel, and the limited external validity of using 2 cell lines, one for certain key experiments. Additionally, to make the stated conclusions, there should be a dose-response analysis.
Major issues:
- The impact on phenotypes on target cells of the altered EVs are not demonstrated—particularly important as some of the effects on EVs are statistically significant but biologically modest in % effect (e.g. Fig 4G or 5B).
- Some of the data are somewhat discordant, such as that paclitaxel reduces CD63+ sEVs in both cell lines (Figure 3C), without affecting distribution of CD63+ fluorescent nanoparticles. How can this be explained?
- Only 2 cell lines are used and only one in Figure 6 for the key ILK experiments one HRD cell line is used, making the data less generalizable. Is it possible to find another ILK-expressing cell or is this rare in TNBC. If rare, then the conclusions should be moderated because ILK as a EV cargo would be a rather infrequent occurrence.
- Effects of most drugs are concentration dependent, and only one concentration is used for each. So this tends to undermine the conclusion that the effects are drug dependent, because they could be simply dose dependent.
- Figure 6A is missing loading controls and molecular weight markers.
Minor:
- P2 line 69 “mitosis is not the major mechanism of action” references review articles from around 2011-12. More recent publications, based on direct sampling of patient tumors in humans and mice have changed this view. (PMIDs 24670687, 30429313)—these studies suggest mitosis is altered without mitotic-arrest, leading to cell micronucleation; the second one is from Mitchison who authored original cite 25.
- Figure 1A needs scale bars.
- Figure 1B-C show representative examples of the region masks.
- What was the rationale for using 500 nM PTX in figure 1 and 50 nM in the subsequent figures? Agree that 50 nM is more likely the physiologic dose
- P6 line 238 –the plasma peak concentration is not very useful to estimate intratumoral levels. The PMID 24670687 above measured about 1 uM intratumoral level of paclitaxel at 20h, but ~200x intracellular concentration, supporting 50nM as a cell culture concentration that mimics pharmacologic level.
- Figure 5 would have been better by including eribulin as the positive control so it is possible to judge the difference in the same experiment. Further, colchicine or nocodazole would have further bolstered the conclusions.
Author Response
Reviewer 1.
The strengths of this manuscript is it directly compares the effects of different MTAs on EVs and their contents, using distinct methods including immunoblotting, EM, and sEV using Alpha Nano Tech instruments.
We appreciate the reviewer’s comments.
The data shown largely support the stated conclusions, with the caveats. Some of the immunoblots, particularly in Figure 3, appear very well done. As noted below, the principle weaknesses concern the lack of demonstrated biological impact of the EV alterations, the discordant data for paclitaxel, and the limited external validity of using 2 cell lines, one for certain key experiments. Additionally, to make the stated conclusions, there should be a dose-response analysis.
These concerns are addressed in detail below.
Major issues:
- The impact on phenotypes on target cells of the altered EVs are not demonstrated—particularly important as some of the effects on EVs are statistically significant but biologically modest in % effect (e.g. Fig 4G or 5B).
We agree that the ultimate biological impact of the altered EVs on recipient cells was not demonstrated, and while the changes we measured are statistically significant, whether they are biologically significant will require many additional experiments. At this point we can hypothesize a potential impact based on the work of others. We believe that our data provide an important first proof-of-principle for future investigations that will establish the biological and clinical significance of the effects of different MTA-induced alterations to EVs. These will be critical and important follow-up experiments for another manuscript.
- Some of the data are somewhat discordant, such as that paclitaxel reduces CD63+ sEVs in both cell lines (Figure 3C), without affecting distribution of CD63+ fluorescent nanoparticles. How can this be explained?
This is an excellent point and we have added new language in the discussion to better explain how we think this could be the case. We think this largely stems from the technical differences between western blotting and fluorescent NTA (fNTA). While western blots can show the amount of CD63 in a sample on a continuum, fNTA is a binary measure and looks at individual particles, telling us simply “yes or no” as to whether they contain CD63. With fNTA there is a threshold amount of CD63 on an EV particle that will trigger detection using a fluorescent antibody. We propose that in the paclitaxel EV samples that while the amount of CD63 packaged per EV particle is reduced and reflected in lower CD63 in the sample by western blotting, this is still above the threshold for “yes” as detection by fNTA. These data show the value of multiple types of analyses.
- Only 2 cell lines are used and only one in Figure 6 for the key ILK experiments one HRD cell line is used, making the data less generalizable. Is it possible to find another ILK-expressing cell or is this rare in TNBC. If rare, then the conclusions should be moderated because ILK as a EV cargo would be a rather infrequent occurrence.
As we show in Figure S10 (previously S9), ILK is highly expressed in MDA-MB-231 cells, however, in contrast to HCC1937 cells, extremely low levels of ILK are detected in the EVs from MDA-MB-231 cells. Thus, the cellular levels do not predict EV levels. We understand the reviewer’s concern about the generalizability of these findings, and thus future studies are necessary to identify other cell lines where high levels of ILK are packaged in EVs, and whether it is detectable in EVs from patients’ serum. These extensive experiments are natural and important follow-up studies. We have added new language in the Discussion to address this caveat, “Future studies should identify additional TNBC cell lines where ILK is highly expressed in sEVs and if it is altered by MTAs. Important follow-up studies should evaluate the effects of MTAs on ILK levels in sEVs derived from patient serum.”
- Effects of most drugs are concentration dependent, and only one concentration is used for each. So this tends to undermine the conclusion that the effects are drug dependent, because they could be simply dose dependent.
We agree that the effects of drugs and MTAs are concentration dependent. In prior studies, that are now referenced, we conducted extensive evaluations of the concentration dependent effects of these MTAs on microtubules, identifying concentrations that maximally disrupt cellular microtubule structure, the target of these drugs, over short time courses, 2-4 hours. Thus, for Figure 1 we had significant prior studies and we evaluated 2 concentrations and saw the same effects on both microtubules and CD63 localization. We show the effects at the lower concentration. For the EV isolation studies again, extensive concentration and time studies were conducted to identify a concentration of paclitaxel and eribulin that affected microtubules and CD63 localization, and yet prevented significant cell death and cell cycle arrest over a time course that yielded sufficient EVs for analyses. The concentration dependent effects of eribulin and paclitaxel on CD63 localization and microtubule structures at 8 hours is shown in Figure S4 (previously S3). We conducted these important studies in this context to identify a concentration that maximally disrupted both microtubules and CD63 localization to test the hypothesis that this would result in altered EV release and cargo.
- Figure 6A is missing loading controls and molecular weight markers.
We added molecular weight markers to Figure 6A and added the loading controls of GAPDH for the cell lysates, and flotillin-1 for the EVs.
Minor:
- P2 line 69 “mitosis is not the major mechanism of action” references review articles from around 2011-12. More recent publications, based on direct sampling of patient tumors in humans and mice have changed this view. (PMIDs 24670687, 30429313)—these studies suggest mitosis is altered without mitotic-arrest, leading to cell micronucleation; the second one is from Mitchison who authored original cite 25.
We acknowledge that these studies do offer a different yet valid view on the role of mitosis in the efficacy of MTAs. Therefore, we tempered our language in the introduction to say that the non-mitotic effects of MTAs also contribute significantly to the clinical efficacy of these drugs and have added some additional citations to back this up.
- Figure 1A needs scale bars.
Scale bars were added to the merged images in Figure 1A.
- Figure 1B-C show representative examples of the region masks.
An additional Supplementary Figure (S2) is provided which shows representative images for Figure 1B and C. Additionally, diagrams describing the regions and cell spots analyses are provided in this new figure.
- What was the rationale for using 500 nM PTX in figure 1 and 50 nM in the subsequent figures? Agree that 50 nM is more likely the physiologic dose.
The higher concentration of paclitaxel was chosen for the short term, 2 to 4-hour experiments where the aim was to maximally disrupt microtubules to observe if there was any effect on CD63 localization. For subsequent experiments where EVs were being isolated, longer incubation times were required, 8 h, so we determined experimentally (Figure S4) that the lower concentration of 50 nM was still sufficient to maximally disrupt microtubule and CD63 localization over this 8 h time period while limiting cell death and mitotic accumulation.
- P6 line 238 –the plasma peak concentration is not very useful to estimate intratumoral levels. The PMID 24670687 above measured about 1 uM intratumoral level of paclitaxel at 20h, but ~200x intracellular concentration, supporting 50nM as a cell culture concentration that mimics pharmacologic level.
We agree that intratumoral levels of paclitaxel are not equal to plasma concentration due to the high degree of drug concentration within cells. Our goal with this statement was simply to demonstrate that the concentrations used in the experiments are within the range of what could be seen in patients.
- Figure 5 would have been better by including eribulin as the positive control so it is possible to judge the difference in the same experiment. Further, colchicine or nocodazole would have further bolstered the conclusions.
We agree that there is a benefit it to comparing vinorelbine and eribulin side-by- side with eribulin as a positive control, particularly if the effects of vinorelbine had been different than what was observed with eribulin, which was not the case. Regarding the second part of the reviewer’s comment, we stand by our decision to investigate vinorelbine rather than nocodazole or colchicine because, it is clinically relevant, used in the treatment of breast cancer, whereas colchicine and nocodazole are not clinically relevant in the treatment of cancer.

Reviewer 2 Report
The manuscript authored by Pederson et al describes the impact of paclitaxel and eribulin on the EVs from triple negative breast cancer.
1) Introduction: authors fail to highlight the aim of the paper and the added value for the EV-scientific community
2) The introduction lacks of a proper description of EVs
3) Materials and Methods: Chapter 2.2 the amount of plated cells is missing. The same applies for chapter 2.3. Also from where the authors got PTX and eribulin. What was the used concentration? How the authors choose the dosing schedule?
4) For how long sEVs have been stored at minus 80? Freeze and thaw should be avoided when working with EVs
5)Figure 2: scale bar is not visible
6)How the authors explain that sEVs cargo could be altered according to different specific MTAs? What is the mechanism behind this? Is this applied only on breast cancer?
Author Response
Reviewer 2.
- Introduction: authors fail to highlight the aim of the paper and the added value for the EV-scientific community.
We addressed the aim of the paper in the final paragraph of the introduction, but in response to this concern, we have added new language in the first sentence to make it clearer. We have also added a sentence to the end of the paragraph to emphasize the added value of our study to the field of EV and tumor microenvironment research:
“Since EVs from cancer cells can impact oncogenesis and tumor progression and because the biogenesis of EVs is microtubule dependent, the goal of this study was to evaluate if eribulin, or paclitaxel alter small EV (sEV) secretion and cargo. The results show that, indeed, disruption of cellular microtubules by MTAs used in the treatment of breast cancer affects the biogenesis and secretion of sEVs as well as specific sEV cargos, including ILK, with differences noted between microtubule stabilizers and microtubule destabilizers. Importantly, this work adds significant insights to our understanding of the complex effects of these cancer therapeutics on EVs and the tumor microenvironment.”
2) The introduction lacks of a proper description of EVs.
We have added a new paragraph to the introduction (paragraph 2) that describes EVs and their subtypes in more detail.
3) Materials and Methods: Chapter 2.2 the amount of plated cells is missing. The same applies for chapter 2.3. Also from where the authors got PTX and eribulin. What was the used concentration? How the authors choose the dosing schedule?
We have added the cell numbers plated for the experiments in sections 2.2 and 2.3.
The sources of paclitaxel and eribulin are provided in the last sentence of section 2.1. Paclitaxel was obtained from Sigma Aldrich and eribulin from Eisai Inc.
The concentrations of drugs used for each experiment are provided in each figure legend. There is no dosing schedule used in this study. We believe that the authors are referring to the choice of time and concentration for the experiments. The rationale behind the choices of concentration and duration of treatment for experiments where sEVs were isolated is stated in section 3.2. In section 3.1 we have added a sentence which references prior studies supporting our choice of short time courses and higher concentrations for the CD63 immunofluorescence experiments.
4) For how long sEVs have been stored at minus 80? Freeze and thaw should be avoided when working with EVs.
This is an important point and we thank the reviewer for bringing it to our attention. We have added the following statements to the methods section. EVs were stored at -80 for no more than 4 months prior to NTA or western blot analysis. For all NTA analyses, only one freeze thaw cycle was allowed, for western blots no more than about 3 freeze thaw cycles were permitted.
5) Figure 2: scale bar is not visible.
To make the scale bars more visible we put the scale number above the bars so that they would stand out more in the figure.
6) How the authors explain that sEVs cargo could be altered according to different specific MTAs? What is the mechanism behind this? Is this applied only on breast cancer?
This is an important question that we have addressed in the discussion. While eribulin and paclitaxel are both MTAs, they differ greatly in how they affect the structure of cellular microtubules. While eribulin causes a loss of microtubules, paclitaxel causes an increased density of microtubules and microtubule bundles. As we say in the first paragraph of the discussion: “differences in the effects of eribulin and vinorelbine compared to paclitaxel, which caused a strikingly different pattern of CD63 staining as defined by spot number, intensity and area suggest that microtubule destabilizers and stabilizers disrupt sEV biogenesis pathways differently. While MTAs are often considered a single class of drugs, these results highlight differences between destabilizers and stabilizers and are consistent with their divergent effects on intracellular signaling including inhibition of TGF-β-induced nuclear translocation of Smad2/3 [36], and mitochondrial biogenesis and polarization[54,55].” In a newly added paragraph (#5) in the discussion we reiterate this by saying “Ultimately, these data suggest that the different effects of microtubule depolymerizers and stabilizers on the intracellular localization of CD63 also leads to diverging effects on CD63 loading into sEVs.”
Regarding the applicability of our findings beyond breast cancer we would hypothesize that because microtubules are the key mediators of intracellular protein and vesicle trafficking in all cell types, differences in the effects of microtubule stabilizers and destabilizers on processes dependent on this trafficking would be expected.

Round 2
Reviewer 2 Report
I suggest the MS for publication